# Associated-Extraction Efficiency of Six Cyclodextrins on Various Flavonoids in Puerariae Lobatae Radix

**DOI:** 10.3390/molecules24010093

**Published:** 2018-12-27

**Authors:** Tao Feng, Fan Liu, Lili Sun, Hongna Huo, Xiaoliang Ren, Meng Wang

**Affiliations:** 1School of Chinese Materia Medica, Tianjin University of Traditional Chinese Medicine, Tianjin 301617, China; mvpfengtao@163.com (T.F.); 15895898263@163.com (F.L.); 18322050489@163.com (L.S.); 15100273885@163.com (H.H.); renxiaoliang@tjutcm.edu.cn (X.R.); 2Tianjin State Key Laboratory of Modern Chinese Medicine, Tianjin University of Traditional Chinese Medicine, Tianjin 301617, China

**Keywords:** Puerariae Lobatae Radix, flavonoids, cyclodextrins

## Abstract

Puerariae Lobatae Radix (PLR), a well-known herbal medicine, is the root of *Pueraria lobata* (Willd.) Ohwi and has been employed for the treatment and prevention of cardiovascular and cerebrovascular diseases. The purpose of this study was to compare the associated-extraction efficiency of six cyclodextrins (CDs) on five flavonoids in PLR, namely puerarin, daidzein, daidzin, genistein and genistin, which are the major secondary metabolites, and exhibit low water solubility. The six CDs applied were β-cyclodextrin (β-CD), γ-cyclodextrin (γ-CD), hydroxypropyl-β-cyclodextrin (HP-β-CD), hydroxypropyl-γ-cyclodextrin (HP-γ-CD), carboxymethyl-β-cyclodextrin (CM-β-CD), and sulfobutyl ether β-cyclodextrin (SBE-β-CD). They can be grouped into one of the following three categories: traditional cyclodextrins (β-CD and γ-CD), water-soluble cyclodextrin derivatives (HP-β-CD and HP-γ-CD) and ionic cyclodextrin derivatives (SBE-β-CD and CM-β-CD). High-performance liquid chromatography (HPLC) was used to analyze the five flavonoids in the original aqueous extracts (OAE) in the presence or absence of various CDs. The associated-extraction efficiency of the various CDs followed the ranking: SBE-β-CD > HP-β-CD > CM-β-CD > HP-γ-CD > γ-CD > β-CD. It was clear that SBE-β-CD presented the highest associated-extraction capability, and it was used to extract the four flavonoids from three PLR products, including raw product, stir- fried product, and product simmered with wheat bran. The results showed that SBE-β-CD could improve the extraction capability of flavonoids, both from the raw product and in processed products of PLR. In conclusion, CDs, especially SBE-β-CD, have a promising application for the associated-extraction of flavonoids from PLR.

## 1. Introduction

Puerariae Lobatae Radix (PLR) is the root of *Pueraria lobata* (Willd.) Ohwi and is a traditional Chinese herbal medicine used to treat colds, high blood pressure and diabetes. A large number of flavonoids in PLR are the main active ingredients, including flavonoid glycosides (puerarin, daidzin and genistin) and flavonoid aglycones (daidzein and genistein) [1]. These secondary metabolites have many beneficial physiological effects including antipyretic and anti-inflammatory actions, dilation of the blood vessels, lowering of blood pressure, and anti-oxidant and anti-cancer effects [2,3,4]. PLR is a dual-use resource for medicine and food, and it is hoped to increase the bioavailability of the drug by extracting and concentrating the main components. The solubility of flavonoids in water is poor, and water solubility is therefore one of the important factors affecting their bioavailability. Only by enhancing the water solubility of the flavonoids can one effectively expand their application, thereby improving their bioavailability. In recent years, various methods for extracting flavonoids have been developed, such as organic solvent extraction, ultrasound-assisted extraction, microwave extraction, and supercritical fluid extraction [5,6,7].

Cyclodextrins (CDs) are well-known cyclic oligosaccharides composed of α-(1,4)- linked glucose units and are widely used because of their “internal-external” affinity, which is mainly due to its special internal hydrophobic and external hydrophilic molecular structure [8]. They can be grouped into the following three categories: traditional cyclodextrins, water-soluble cyclodextrin derivatives and ionic cyclodextrin derivatives [9]. They have particular internal hydrophobic and external hydrophilic molecular structures, which can be combined with poorly soluble drug molecules to form complexes, increasing the solubility of the drug [10]. CDs are considered to be promising pharmaceutical excipients that form water-soluble inclusion complexes with various poorly soluble compounds [11]. They can improve the extraction efficiency of Chinese herbal medicinal products, increasing the bioavailability of the drug [12]. A variety of Chinese herbal medicinal extracts can form complexes with CDs, thereby improving the extraction efficiency and bioavailability of the active ingredients. Studies have shown that CDs can be used for the extraction of various components such as phenolic compounds and anthocyanins [13,14,15,16]. Being starch derivatives, CDs are non-toxic and non-polluting. They are safer and more promising extractants than organic solvents.

The traditional extraction methods for flavonoids have low extraction efficiencies. At present, the newly emerging extraction methods generally need to use organic solvents, which are prone to causing environmental pollution, or which require high large amounts of equipment and entail high energy consumption and high cost. Therefore, these new extraction methods are widely used in scientific research but has not so widely used in the practice of the extraction of natural compounds. Given that CDs have good extraction activity, low requirements for extraction equipment, and require avoiding the use of toxic organic solvents, they are more suitable for the production of herbal extractions. In our previous studies, we have studied the extraction of various components of the traditional Chinese formula Xue Zhi Ning by CDs and the associated-extraction efficiency of flavonoids [17,18]. In the current study, the associated-extraction efficiency of flavonoids from PLR by different CDs was analyzed by high-performance liquid chromatography (HPLC). The associated-extraction efficiencies of different classes of CDs were analyzed by principal component analysis (PCA) and counter-propagation artificial neural networks (CP-ANN). Finally, we used the CD with the strongest extraction ability (SBE-β-CD) to extract the PLR raw products and different processed products, to compare the associated-extraction efficiency by SBE-β-CD of these products, relative to that of conventional extraction processes. This study should provide a reference for solving the problem of the low water solubility of the flavonoids in PLR.

## 2. Experiment

### 2.1. Chinese Herbal Medicines and Reagents

Three products of PLR, namely the raw product, the stir-fried product, and the product simmered with wheat bran, were purchased from Bozhou Jingyu Traditional Chinese Medicine Pieces Factory. (Bozhou, Anhui, China). β-Cyclodextrin, γ-cyclodextrin, hydroxypropyl-β-cyclodextrin, hydroxypropyl-γ-cyclodextrin, carboxymethyl-β-cyclodextrin, and sulfobutyl ether β-cyclodextrin were purchased from Shandong Binzhou Zhiyuan Biotechnology Co., Ltd. (Binzhou, Shandong, China). Puerarin, daidzein, daidzin, genistein and genistin (all purities > 98%) were purchased from Weikeqi Biological Technology Co., Ltd. (Chengdou, Sichuan, China).

### 2.2. Preparation of Samples

The raw product of PLR (5 g) and individual CDs (0.25 g) were combined in water (125 mL). All samples were soaked for one hour, and then decocted for two hours. The extracted solution of each CD was obtained and its volume was recorded. The preparation process for the original aqueous extract (OAE) was the same as described above, but without the addition of any CD. For HPLC analysis, an aliquot (1 mL) of different extracts were diluted with methanol at different multiples according to the content of the flavonoids. All samples were exposed to ultrasound for 30 min at 25 °C and then centrifuged at 11,750× *g* (TG16-WS Centrifuge, Hunanxiangyi Laboratory Instrument Development Co., Ltd., Changsha, China) for 10 min. Before injection, all the solutions were filtered through a 0.22 µm nylon membrane.

### 2.3. HPLC Analysis

Chromatographic analyses were performed using an LC-20AT HPLC system (Shimadzu, Kyoto, Japan), equipped with a single column heater, a binary solvent manager, a sample manager and a diode-array detector. All separations were carried out on an Elite SinoChrom ODS-BP column (250 mm × 4.6 mm, 5 μm i.d.; Dalian Elite Analytical Instrument Co., Ltd., Dalian, China) maintained at 40 °C. The mobile phase consisted of formic acid/water (1 mL/L) and methanol (solvent B). The gradient program was developed as follows: 5–95% B at 0–40 min, 95% B at 40 min, with an analytical time of 8 min. The injection volume was 10 μL and the flow rate was 1 mL/min. The wavelength of the detector was set at 254 nm (for puerarin, daidzein, daidzin and genistin) or at 360 nm (for genistein) and the peak areas under the curve were recorded.

### 2.4. Statistical Analysis

Data were processed using IBM SPSS 19.0 software (IBM Corp., Armonk, NY, USA); the summary statistics were expressed as mean ± standard deviation. Any two means were compared using Student’s *t*-test. *p* < 0.05 was deemed to be statistically significant. PCA analysis was performed using simca-p14.1 (Umetrics, Umeå, Sweden); CP-ANN was derived from MatlabR2014a (MathWorks, Natick, MA, USA) with Kohonen and CP-ANN toolbox 3.6 (Milano Chemometrics and QSAR Research Group, Milano, Italy).

#### 2.4.1. Evaluation of Associated-Extraction Efficiency Using HPLC Analysis

The extraction growth rate can be calculated by the following equation
Extraction growth rate = (A_1_ − A_0_)/A_0_ × 100%
A_1_ and A_0_ represent the peak area of each flavonoid in the CDs and OAE, respectively. The peak area of each flavonoid in the OAE was scored as 1 point. The peak area of each flavonoid in the CD was compared with 1, and the extraction growth rate was obtained. The higher the score, the greater the associated-extraction efficiency.

#### 2.4.2. Principal Component Analysis (PCA)

PCA is a data reduction algorithm that generates a set of principal components that are independent linear combinations of the original dataset [19,20,21]. These principal components can be used to analyze the similarity of sample clusters and relate information between load variables and samples [22]. Through PCA analysis, rapid cluster analysis of samples can be achieved, and samples with the same or similar characteristics can be grouped together to find common features. PCA is a commonly used statistical analysis method for removing abnormal samples. Circles represent 95% confidence intervals. Samples outside the circle represent outliers [23]. In this study, PCA was applied to compare the associated-extraction efficiency of different CDs for the flavonoids in PLR.

#### 2.4.3. Counter-Propagation Artificial Neural Network (CP-ANN)

The Kohonen network is a self-organizing competitive neural network that recognizes environmental features and automatically clusters them. CP-ANN is based on the Kohonen network and is capable of handling unsupervised and supervised classification problems. It consists of two layers: the Kohonen layer and the output layer [24,25]. After entering the data, the neurons in the Kohonen layer respond to different patterns of the sample; the samples are assigned to neurons in different regions with different patterns [26]. In this study, PCA was applied to compare and visualize the associated-extraction efficiency of the five flavonoids by CDs.

## 3. Results and Discussion

### 3.1. Evaluation of Associated-Extraction Efficiency Using HPLC Analysis

#### 3.1.1. Associated-Extraction Efficiency of Five PRL Flavonoids by Six CDs

The peak areas results of the five flavonoids are shown in Table 1, they represent the associated-extraction efficiencies of the six CDs for the flavonoids. Compared with OAE, the peak area for each flavonoid was significantly higher after using CDs, indicating the increased associated-extraction efficiency of the CDs. Table 2 shows the results for the extraction growth rate of the flavonoids in PLR, the extraction growth rate of the OAE was lower than that of the CDs derivatives. The peak area of each flavonoid without the addition of the CDs was scored as 1 point. The peak area of each flavonoid in the different CD solutions was compared with 1, and the extraction growth rate was obtained. The higher the score, the greater the solubility of the flavonoid in water. The results are shown in Table 3. The radar map for the individual flavonoids is shown in Figure 1. The radar map demonstrates the ordering of the six CD derivatives on the solubilization of flavonoids from PLR: SBE-β-CD > HP-β-CD > CM-β-CD > HP-γ-CD > γ-CD > β-CD. SBE-β-CD had the greatest solubilizing effect for genistein and daidzin, while HP-γ-CD had the greatest solubilizing effect for daidzein, and HP-β-CD showed the greatest solubilization of puerarin and genistin.

#### 3.1.2. Principal Component Analysis (PCA)

PCA was used to visualize the associated-extraction efficiency of the different CDs. In the PCA model, the cumulative contribution rate (R2X(cum) = 0.832) and cross validation (Q2(cum) = 0.514) indicate that the model has good analysis and prediction capabilities. The result is shown in Figure 2. It can be seen that the samples are grouped into four clusters: the OAE are grouped together into one cluster (class 1), and the samples of β-CD and γ-CD are clustered into a separate cluster (class 2). The samples to which HP-β-CD and HP-γ-CD were added were grouped together into the same cluster (class 3), while the samples to which CM-β-CD and SBE-β-CD were added were grouped together into the final cluster (class 4). The classification results are related to the molecular structures: β-CD and γ-CD are traditional CDs, HP-β-CD and HP-γ-CD are water-soluble CD derivatives, and CM-β-CD and SBE-β-CD are ionic CD derivatives. The results of the analysis show that the CD derivatives had different effects on extraction of the flavonoids, while the same type of CD derivatives had similar effects. The type of CD derivative was the major factor affecting the solubility of the flavonoids. According to the established PCA model, PLR with different CD derivatives can be clearly distinguished.

#### 3.1.3. Counter-Propagation Artificial Neural Networks (CP-ANN)

The results of the CP-ANN analysis are shown in Figure 3. Neurons of different colors represent different types, while neurons of the same color have similar distributions. Different types of CD-treated samples occupy different neurons, the same type of CD-treated sample occupies the same or similar neurons, and there is no intersection between different types of neuron, indicating that the experimental results were good. Through the CP-ANN analysis, the PCA results can be validated to prove that different types of CDs have different effects on the extraction of flavonoids. The Kohonen weights were obtained from the CP-ANN variables, and the associated-extraction efficiency of different types of CD was measured using a limit of 0.35. The results are shown in Figure 4: the three types of CD have obvious effects on flavonoids. The effects of class 3 and class 4 CDs are larger than those in class 2, with the most obvious effects being shown by class 4. Class 4 has associated-extraction efficiency for five different flavonoids, and can significantly enhance their solubility. These findings are consistent with the conclusions drawn from the PCA and peak area analysis. This indicates that the analytical method used during the experiment is appropriate.

### 3.2. Associated-Extraction Efficiency of SBE-β-CD for Flavonoids from Different Processed Products of PLR

SBE-β-CD showed significant associated-extraction efficiency and has obvious effects on flavonoid extraction from different processed Chinese herbal medicinal plants, but the solubilization effect varied (Table 4). The extraction growth rate of flavonoids in various PLR materials is shown in Figure 5. With the exception of daidzein, the remaining three flavonoids had obvious effects on the stir-fried product and simmered product with wheat bran, but the overall gap was not large. SBE-β-CD had a good solubilization effect on the raw product, and the solubility of all flavonoids was greatly improved. On the other hand, the solubilization effect of daidzein from the raw product, stir-fried product and simmered product was better than daidzin, indicating that SBE-β-CD has a greater solubilizing effect on flavonoid aglycones than on flavonoid glycosides, and the solubilization effect on compounds which are usually barely soluble in water was good.

## 4. Conclusions

Different CDs have different solubilization effects on flavonoids; among them, SBE-β-CD has the most obvious effect. By comparing the flavonoid aglycones and flavonoid glycosides, it has been shown that the solubilization effect of CDs for flavonoid aglycones is greater than that for flavonoid glycosides, while the solubilization effect on compounds which are hardly soluble in water is better. In addition, statistical analysis showed that different types of CD have different solubilization effects, and the order of the results were as follows: ionic cyclodextrin derivatives > water-soluble cyclodextrin derivatives > traditional cyclodextrins. SBE-β-CD has no selective solubilization effect on flavonoids in different processed products of PLR and has a good solubilizing effect on flavonoids which are barely soluble in water. However, Chinese medicine extraction is a complex chemical process, subject to various internal and external factors, and different classes of CD have different associated-extraction efficiencies on various individual flavonoids. It is necessary to select a suitable CD according to the specific target.

## Figures and Tables

**Figure 1 molecules-24-00093-f001:**
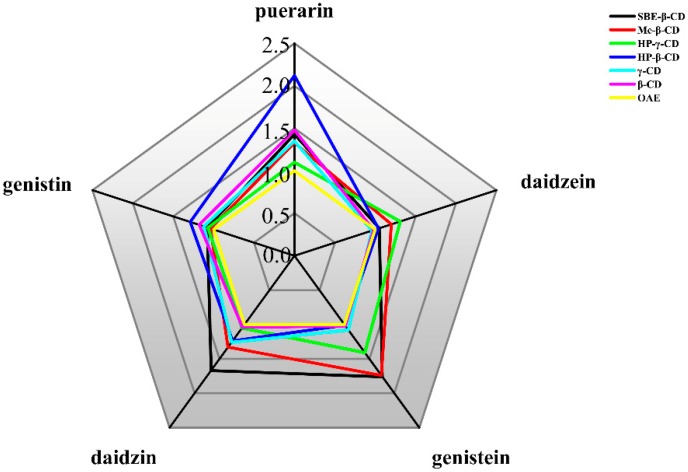
Associated-extraction efficiency radar map of different CDs.

**Figure 2 molecules-24-00093-f002:**
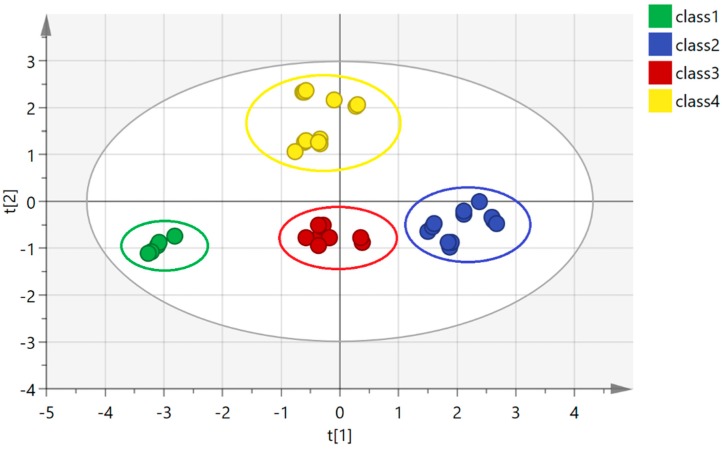
PCA score scatter plot for different classes of CDs.

**Figure 3 molecules-24-00093-f003:**
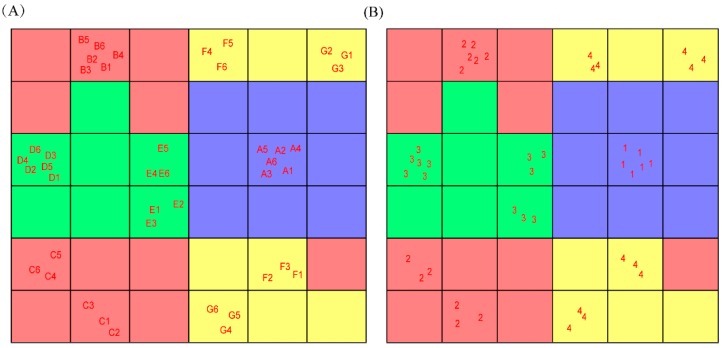
Kohonen map of associated-extraction efficiency of different cyclodextrins. (**A**) The distribution of different cyclodextrins and (**B**) the classes assigned to cyclodextrins (1: OAE, 2: traditional cyclodextrins, 3: water-soluble cyclodextrin derivatives, 4: ionic cyclodextrin derivatives).

**Figure 4 molecules-24-00093-f004:**
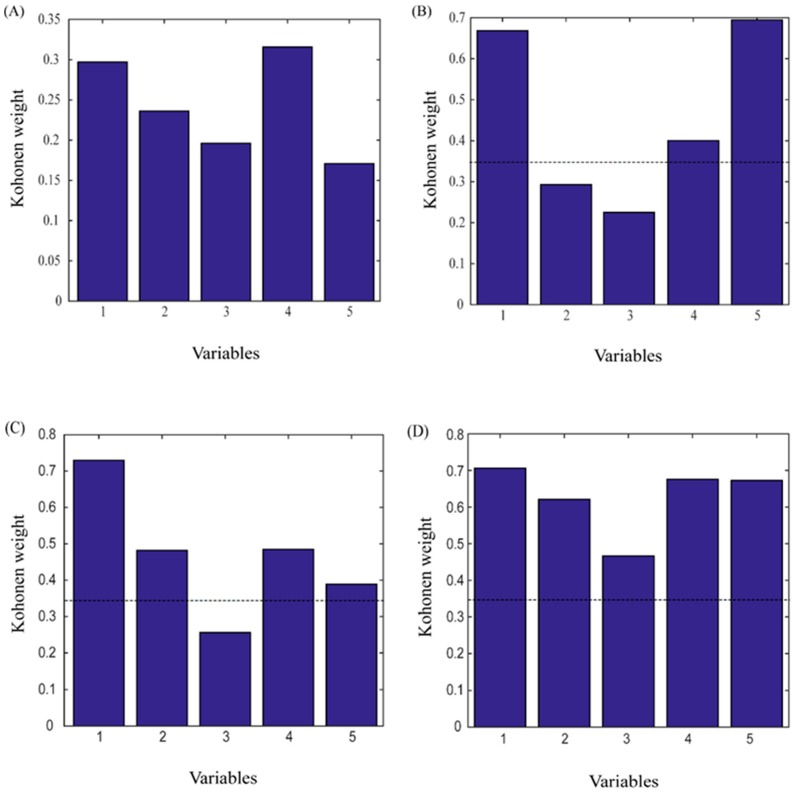
The Kohonen weights of the five flavonoids in four classes of associated-extraction efficiency from CP-ANN. (**A**: OAE, **B**: traditional cyclodextrins, **C**: water-soluble cyclodextrin derivatives, **D**: ionic cyclodextrin derivatives 1: puerarin, 2: daidzein, 3: genistein, 4: genistin, 5: daidzin).

**Figure 5 molecules-24-00093-f005:**
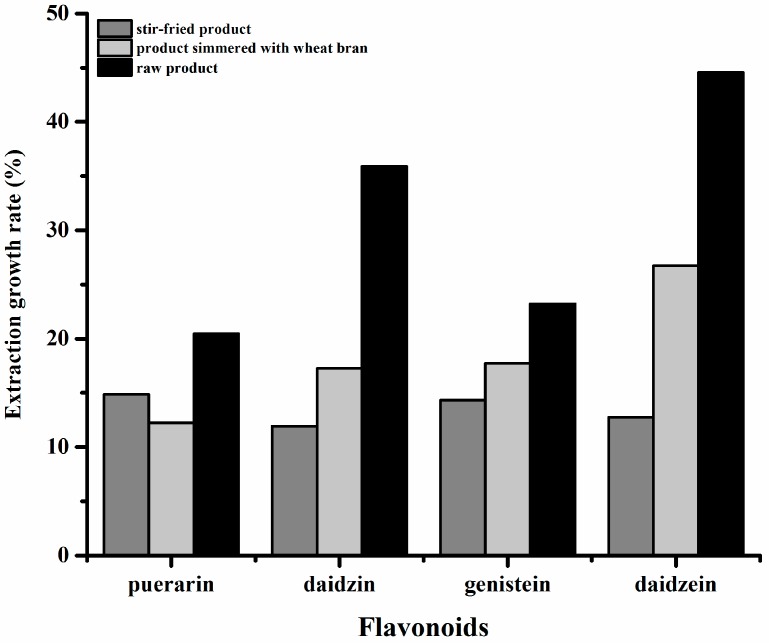
Extraction growth rate of flavonoids in various PLR materials (%).

**Table 1 molecules-24-00093-t001:** Associated-extraction efficiency of six CDs for flavonoids (peak area, *n* = 6).

Groups	Peak Area				
	Puerarin	Daidzein	Genisten	Daidzin	Genistin
sulfobutyl ether β-cyclodextrin	2,029,213 ± 203,886 **	221,198 ± 43,008	36188 ± 10293 **	305,795 ± 51,497 **	648,695 ± 65,923
carboxymethyl-β-cyclodextrin	1,930,712 ± 247,991 **	270,287 ± 28,210 **	27,443 ± 6175	241,245 ± 46,006 *	746,623 ± 94,570 *
hydroxypropyl-γ-cyclodextrin	2,179,303 ± 456,406 **	227,414 ± 67,279	18,593 ± 833	172,658 ± 54,210	668,197 ± 31,088
hydroxypropyl-β-cyclodextrin	3,024,023 ± 836,080 **	219,235 ± 183,98	20,889 ± 3454	226,944 ± 12,877 **	772,266 ± 238,888
γ-cyclodextrin	1,915,263 ± 264,897 *	201,910 ± 46,723	22,348 ± 579	229,948 ± 58,279	669,958 ± 178,231
β-cyclodextrin	2,113,199 ± 58,178 **	206,347 ± 6211	21,097 ± 4635	189,898 ± 43,431	709,756 ± 33,611 *
original aqueous extracts	1,423,886 ± 304,315	210,638 ± 33,799	20,610 ± 5023	183,151 ± 11,279	599,673 ± 81,374

* differences from the original aqueous extract are significant (*p* < 0.05), using Student’s *t*-test. ** differences from the original aqueous extract are significant (*p* < 0.01), using Student’s *t*-test.

**Table 2 molecules-24-00093-t002:** Extraction growth rate of flavonoids in PLR (%).

Groups	Flavonoids				
	Puerarin	Daidzein	Genistein	Daidzin	Genistin
sulfobutyl ether β-cyclodextrin	42.51	5.01	75.59	66.96	8.17
carboxymethyl-β-cyclodextrin	35.59	28.32	33.16	31.72	24.50
hydroxypropyl-γ-cyclodextrin	10.03	31.37	40.78	4.51	7.43
hydroxypropyl-β-cyclodextrin	112.38	4.08	1.36	23.91	28.78
γ-cyclodextrin	34.51	−4.14	8.43	25.55	11.72
β-cyclodextrin	48.41	−2.04	2.36	3.68	18.36

**Table 3 molecules-24-00093-t003:** Associated-extraction efficiency of different CDs (score).

Groups	Flavonoids					Sum
	Puerarin	Daidzein	Genistein	Daidzin	Genistin	
sulfobutyl ether β-cyclodextrin	1.43	1.05	1.76	1.67	1.08	6.98
carboxymethyl-β-cyclodextrin	1.36	1.28	1.33	1.32	1.25	6.53
hydroxypropyl-γ-cyclodextrin	1.10	1.31	1.41	1.05	1.07	5.94
hydroxypropyl-β-cyclodextrin	2.12	1.04	1.01	1.24	1.29	6.71
γ-cyclodextrin	1.35	0.96	1.08	1.26	1.12	5.76
β-cyclodextrin	1.48	0.98	1.02	1.04	1.18	5.71
original aqueous extracts	1	1	1	1	1	5

**Table 4 molecules-24-00093-t004:** Associated-extraction efficiency of sulfobutyl ether β-cyclodextrin for flavonoids from different processed products of PLR (peak area, *n* = 3).

Medicinal Herbs	Flavonoids	Original Aqueous Extracts	Sulfobutyl Ether β-Cyclodextrin
stir-fried product	Puerarin	4,962,426 ± 123,669	5,700,551 ± 297,235 *
	Daidzin	1,285,994 ± 47,366	1,439,234 ± 11,338
	Genistein	273,012 ± 3154	312,108 ± 8207 *
	Daidzein	200,883 ± 2281	226,512 ± 28,244
product simmered with wheat bran	Puerarin	4,348,354 ± 33,017	4,880,039 ± 158,484 **
	Daidzin	1,330,505 ± 79,428	1560,305 ± 32,950 **
	Genistein	455,096 ± 39,482	535,801 ± 8466 *
	Daidzein	325,621 ± 27,850	412,680 ± 8902 **
raw product	Puerarin	6,678,584 ± 698,346	8,045,093 ± 239,842 *
	Daidzin	1830,292 ± 61,375	2,487,306 ± 150,734
	Genistein	371,929 ± 2880	458,303 ± 7741 **
	Daidzein	274,732 ± 6077	397,165 ± 2822 **

* differences from the original aqueous extract are significant (*p* < 0.05), using Student’s *t*-test. ** differences from the original aqueous extract are significant (*p* < 0.01), using Student’s *t*-test.

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
