# Peer review of "Associated-Extraction Efficiency of Six Cyclodextrins on Various Flavonoids in Puerariae Lobatae Radix"

_molecules, 2018, doi:10.3390/molecules24010093_

Reviewer 1 Report

The manuscript of  Tao Feng and co-workers “Associated-extraction efficiency of six cyclodextrins  on various flavonoids in Puerariae Lobatae Radix” is interesting in the field and I would recommend publishing it after some corrections:

·         Abstract: The Latin name of the plant should be written italic

·         The tables are unreadable. You can not compare the data. Please correct the tables.

·         How did the authors identify the flavonoids. Has the analysis occurred by comparing with the purchased standards?

Author Response

Dear Professor,

  Thank you very much for your comments and advice about our paper submitted to molecules (molecules-410004). We have learned much from the comments, which are fair, encouraging and constructive. After carefully studying the comments and your advice, we have made corresponding changes. We submit here the revised manuscript as well as a list of changes. And the revised portions were marked in red bold in revised manuscript.

1.     Abstract: The Latin name of the plant should be written italic.

Change: The Latin name of the plant have been written italic in the abstract.

2.     The tables are unreadable. You can not compare the data. Please correct the tables.

Change: The format of the tables has been modified in revised manuscript.

3.     How did the authors identify the flavonoids. Has the analysis occurred by comparing with the purchased standards?

Response: Yes. The analysis occurred by comparing with the purchased standards.

Reviewer 2 Report

This article studies the extraction efficiency of six cyclodextrins on various flavonoids in Puerariae Lobatae Radix. According to the literature, the authors are experts on the field and the implications of this new research and the differences with the previous ones can be better highlighted in the introduction. One special aspect requires revision: the number of authors in the MDPI platform (Tao Feng , Fan Liu , Lili Sun , Hongna Huo , Xiaoliang Ren , Meng Wang) is different to the presented in the article itself ( Tao Feng a, Meng Wang b, Xiaoliang Rena,). Other aspects to consider are:

1.      The use of peak areas in some tables does not provide a clear idea of the results. Relative parameters are preferred.

2.      Revise the format of the tables

3.      Include error bars in Figure 5.

Author Response

Dear Professor,

  Thank you very much for your comments and advice about our paper submitted to molecules (molecules-410004). We have learned much from the comments, which are fair, encouraging and constructive. After carefully studying the comments and your advice, we have made corresponding changes. We submit here the revised manuscript as well as a list of changes. And the revised portions were marked in red bold in revised manuscript.

1.   The number of authors in the MDPI platform (Tao Feng , Fan Liu , Lili Sun , Hongna Huo , Xiaoliang Ren , Meng Wang) is different to the presented in the article itself ( Tao Feng a, Meng Wang b, Xiaoliang Rena,).

Change: The number of authors in the MDPI platform (Tao Feng , Fan Liu , Lili Sun , Hongna Huo , Xiaoliang Ren , Meng Wang) is correct, and we have corrected this error in revised manuscript .

2.    The use of peak areas in some tables does not provide a clear idea of the results. Relative parameters are preferred.

Response: The flavonoids have a defined structure, and the measured ingredients have similar absorption energies at the measurement wavelength. We use self-controlled experiments to compare the growth rate of the ingredients.

3.      Revise the format of the tables

Change: The format of the tables has been modified in revised manuscript.

4.      Include error bars in Figure 5.

Response: Figure 5 is on the basis of the average data of Table 4. The increase of the peak area after adding sulfobutyl ether β-cyclodextrin is compared with the peak area of the original aqueous extracts to obtain the extraction growth rate of flavonoids.